# Variability of Tacrolimus Trough Concentration in Liver Transplant Patients: Which Role of Inflammation?

**DOI:** 10.3390/pharmaceutics13111960

**Published:** 2021-11-19

**Authors:** Anaelle Chavant, Xavier Fonrose, Elodie Gautier-Veyret, Marie Noelle Hilleret, Matthieu Roustit, Francoise Stanke-Labesque

**Affiliations:** 1University Grenoble Alpes, HP2 INSERM U1300, 38041 Grenoble, France; AChavant@chu-grenoble.fr (A.C.); egautier@chu-grenoble.fr (E.G.-V.); MRoustit@chu-grenoble.fr (M.R.); 2Laboratory of Pharmacology, Pharmacogenetics and Toxicology, Grenoble Alpes University Hospital, 38043 Grenoble, France; XFonrose@chu-grenoble.fr; 3Department of Hepato-Gastroenterology, Grenoble Alpes University Hospital, 38041 Grenoble, France; MNHilleret@chu-grenoble.fr; 4Clinical Investigation Center, Grenoble Alpes University Hospital, 38043 Grenoble, France

**Keywords:** tacrolimus, trough concentration, variability, inflammation

## Abstract

Tacrolimus presents high intra and inter-individual variability in its blood trough concentration (Cmin). Knowledge of the factors that are involved in tacrolimus Cmin variability is thus clinically important to prevent or limit it. Inflammation can affect the pharmacokinetic properties of drugs. We evaluated the contribution of acute inflammation in the pharmacokinetic variability of tacrolimus blood Cmin in a large cohort of liver transplant patients. Demographic, biological, and clinical data from 248 liver transplant patients treated with tacrolimus from January 2010 to December 2016 were retrospectively collected from medical records. In total, 1573 Cmin/dose and concomitant C-reactive protein (CRP) measurements were analysed. In multivariate analysis, the log Cmin/dose of tacrolimus was significantly and positively associated with the hematocrit, ALAT, and CRP concentrations. CRP concentrations were higher (*p* = 0.003) for patients with tacrolimus overexposure (i.e., tacrolimus Cmin > 15 µg/L) (median CRP (10th–90th percentiles): 27 mg/L (3–149 mg/L), *n* = 91) than they were for patients with a tacrolimus Cmin ≤ 15 µg/L (13 mg/mL (3–95 mg/L), *n* = 1482)). CRP in the fourth quartile (49 to 334 mg/L) was associated with a 2.6-fold increased risk of tacrolimus Cmin overexposure. Our study provides evidence that inflammation contributes to tacrolimus Cmin variability and suggests that inflammation should be considered for the correct interpretation of tacrolimus blood concentration.

## 1. Introduction

Tacrolimus is the most widely used immunosuppressant drug that is used to prevent organ graft rejection after transplantation, but it has a narrow therapeutic window. Thus, therapeutic drug monitoring (TDM) for subsequent dose adjustment is recommended [1].

Tacrolimus presents high inter- and intra-individual variability of its blood trough concentration (Cmin) [2,3]. Moreover, high within-variability of tacrolimus Cmin is a surrogated biomarker of allograft rejection [4,5,6,7,8,9]. For adult liver transplant recipients, long-term tacrolimus blood Cmin variability has been shown to be associated with long-term patient survival [10]. Thus, the identification of the factors that are involved in tacrolimus Cmin variability is clinically important to prevent or limit tacrolimus Cmin variability. 

Individual demographic characteristics, such as age, gender, ethnicity, or poor compliance are known to influence tacrolimus Cmin [11], as are genetic polymorphisms of CYP3A4/5 [12], co-medication with a 3A4/5 inducer or inhibitor [13], or liver dysfunction [14] due to the extensive cytochrome P450 3A4/5-dependent pathway of tacrolimus metabolism [15].

Moreover, recent studies have suggested that acute episodes of inflammation that are related to clinical infectious events [16] or endoscopic retrograde cholangiopancreatography [17] may also contribute to tacrolimus pharmacokinetic variability in liver transplant recipients. Indeed, inflammation can down-regulate certain drug-metabolizing enzymes and transporters [18] or can change the binding of drugs to plasma proteins [19]. The unbound fraction of tacrolimus in the plasma, hematocrit, and serum albumin concentration have also been described as covariates of tacrolimus Cmin [20,21]. However, the influence of inflammation biomarkers on the variability of tacrolimus Cmin has only been evaluated in 10% of tacrolimus pharmacokinetic studies [22]. 

The first aim of this study was to evaluate the contribution of inflammation in the pharmacokinetic variability of the tacrolimus Cmin in the blood in a large cohort of liver transplant patients. A tacrolimus Cmin > 15 µg/L should be avoided [1] to prevent toxicity. Thus, our second objective was to determine whether tacrolimus overexposure (Cmin > 15 µg/L) is associated with an enhanced inflammatory status. 

## 2. Materials and Methods

### 2.1. Study Design

We performed a retrospective monocentric cohort study that was approved by the Grenoble University Hospital review board (registration RnIPH 2020, protocol TACINF; CNIL number: 2205066 v 0). The study was conducted according to the guidelines of the Declaration of Helsinki and was approved by the Institutional Review Board on 26 May 2021 by the CECIC Rhône-Alpes-Auvergne, Clermont-Ferrand (IRB number 5891). Patient consent was waived due to the retrospective design of the study.

In total, 316 patients who had received a liver graft between January 2010 and December 2016 and who had received tacrolimus for the prevention of transplant rejection were eligible (see flow chart). All of the patients also received corticosteroids and mycophenolate mofetil for the prevention of graft rejection.

The inclusion criteria were adult liver transplant patients between 2010 and 2016 and who had been newly hospitalized at the Grenoble University Hospital at least 15 days after their first post-transplant hospitalization or in out-patient follow up, treated with oral tacrolimus formulations (immediate release, Prograf (Astellas) or delayed relase, Advagraf (Astellas) or Envarsus (Chiesi)), and for whom the tacrolimus Cmin was measured concomitantly with their CRP level (±24 h).

The exclusion criteria were patients in their first hospitalization for their liver transplant, as tacrolimus Cmin is highly variable in the early post-transplantation period, which is partially due to ressucitation and the gradual recovery of liver function [7].

Demographic, clinical, biological (CRP, alanine aminotrasferase (ALAT), aspartate aminotransferase (ASAT), hematocrit, bilirubin, tacrolimus Cmin), and pharmaceutical data (tacrolimus daily dose, route of administration) were retrospectively collected from electronic medical records. 

Tacrolimus TDM was performed at a pharmacokinetic steady state by means of the liquid chromatography tandem mass spectrometry method (LC-MS) on blood samples that had been collected just before subsequent tacrolimus administration. 

The inflammatory status was assessed by the CRP concentration.

### 2.2. Analytical Methods

#### 2.2.1. Tacrolimus Quantification by LC-MS

Sample preparation consisted of precipitating 100 µL of whole blood with 200 µL of methanol/0.2M ZnSO_4_ (80/20, *v/v*) containing the internal standard (IS) ^13^C-D_2_-tacrolimus. Samples were vortexed immediately for 30 s. The mixture was centrifuged for 10 min at 25,000× *g*. An amount of 200 µL of supernatant was transferred to integrated microinsert polypropylene HPLC vials. The LC system consisted of two Shimadzu series Prominence LC 20AD quaternary pumps that had been equipped with a Prominence SIL 20AC 70-vials autosampler (operated at 4 °C) and a Shimadzu column oven Prominence CTO-20AC. Online sample clean-up was performed on a purification column (Oasis HLB, 25 µm, 2.1 mm × 20 mm, Waters, MA, USA). Chromatographic separation was performed on a phenyl-hexyl analytical column (Phenomenex Luna, 5 µm, phenyl-hexyl, 2 mm × 50 mm, Aschaffenburg, Germany). The operating procedure for the HPLC-integrated online sample clean-up consisted of two steps: First, 50 µL of the deproteinized sample were injected into the system and were transferred onto the Oasis HLB column. Here, the analytes were adsorbed, whereas potentially interfering matrix compounds were washed directly into the waste by means of a mobile phase consisting of water/methanol 90/10 at a flow rate of 2 mL/min. Following this first step, a six-port valve was switched at 1 min. The extract was then eluted in back-flush mode and was transferred to the analytical column (maintained at 60 °C) with methanol/ammonium acetate 15 mM 97/3 (*v*/*v*) and 0.1% formic acid mobile phase at a flow rate of 0.600 mL/min. After this chromatographic step, the valve was switched back to its original configuration for 1.5 to 2.2 min.

MS/MS analyses were performed on an API 3200 QTRAP mass spectrometer (Sciex, Toronto, ON, Canada) equipped with ESI probe on a Turbo V^®^ ion source. The mass spectrometer was operated in positive mode under the following conditions: ESI electrospray voltage: 5500 V; nebulization gasflowrate: 50 psi; turbo heater gasflowrate: 60 psi; turbo heater temperature: 300 °C. The analyses were performed during multiple reaction monitoring (MRM). Two ion transitions were monitored: (M + NH_4_)^+^
*m/z* 821.3/768.3 for tacrolimus and (M + NH_4_)^+^
*m/z* 824.3/771.3 ^13^C-D_2_-tacrolimus. Each monitored transition dwell time was set to 50 ms in order to obtain at least 15 points per peak. Analyst 1.6.3 software (AB Sciex Pte. Ltd., Singapore) was used for data acquisition and processing. Tacrolimus was purchased from Sigma Aldrich Chemicals (St. Quentin Fallavier, France) and ^13^C-D_2_-tacrolimus was purchased from Alsachim (Strasbourg, France). LC-MS-grade methanol (MeOH) was purchased from Sigma Aldrich, and HPLC-grade ammonium acetate and formic acid were provided by Prolabo (Paris, France). Ultrapure water (resistivity ≥ 18.0 MΩ/cm) was obtained using a Milli-Q Plus (Millipore, Molsheim, France). Polypropylene 2-milliliter (mL) centrifuge tubes, 2-mL tubes with 200-microliter (µL) restrictor screw cap vials, and pipette tips were purchased from Eppendorf (Le Pecq, France), Interchim (Montluçon, France), and Gilson (Middletown, WI, USA), respectively. 

The lower limit of quantification of the tacrolimus was 1 µg/L with a between-day coefficient of variation (CV) of 9.64%, and the uper limit of quantification was 30 µg/L (CV = 3.14%). The between-day CV for the low (2.42 µg/L), medium (7.08 µg/L and 14.10 µg/L), and high levels (32.93 µg/L) of the tacrolimus quality controls that were monitored daily were 7.79%, 5.13%, 5.43%, and 6.85%, respectively. 

#### 2.2.2. Hematocrit, CRP, ALAT, ASAT, Bilirubin and Tota Protein Quantification

Hematocrit determination was performed on a XE 5000 (Sysmex, Kobe, Japan) by means of impedencemetry. CRP concentration was measured by nephelometry, ASAT, ALAT, and bilirubin, and the total protein concentrations were measured by means of colorimetric methods on a Vista 1500 (Siemens Haelthineers, Erlangen, Germany). 

### 2.3. Statistical Analysis

The analysis of the determinants of tacrolimus Cmin variability was performed on the tacrolimus Cmin that had been adjusted for the dose (C/D) to account for the influence of the dose adjustments performed during longitudinal TDM. The relationship between C/D (dependent variable) with other variables (age, sex, hematocrit, ALAT, bilirubin, CRP, and post-transplant delay) was tested using linear mixed-effect models and used patients and post-transplant delay as random factors to account for the multiplicity of tacrolimus Cmin and the CRP concentration obtained for the same patient at different times post-transplantation. Multivariate linear mixed-effect analysis was conducted using all factors and covariates (ALAT, bilirubin, CRP as continuous variable or categorical variable) for which a *p*-value lower than 0.15 was found in the univariate analyses. The relationship between tacrolimus overexposure (Cmin > 15 µg /L) and other variables was tested following the same principle. Univariate and multivariate linear mixed-effect regression analyses for the identification of the determinants of tacrolimus trough concentrations > 15 µg/L (*n* = 91) during longitudinal therapeutic drug monitoring were performed. A generalized logistic mixed model using the patients and post-transplant delay as random factors was also used to assess the impact of ALAT, bilirubin, hematocrit, post-transplant delay (expressed as month quartiles), and CRP (expressed as quartiles) on any tacrolimus overexposure. Post-transplant delays and the CRP quartile interaction on tacrolimus overexposure was also tested. ASAT was not included in statistical models, given its collinearity with ALAT.

The Shapiro–Wilks test was used to assess the normality of the distribution of continuous variables, and Levene’s test was used to assess the homogeneity of the variances. Data were log-transformed to satisfy the application conditions of the linear models when they were not normally distributed. All of the statistical tests were performed at the threshold alpha of 0.05. The statistical analyses were performed using Jamovi^®^ (version 1.6, Syndey, Australia).

## 3. Results

### 3.1. Population Characteristics

Our study population consisted of 248 adult liver transplant patients, for whom 1573 Cmin/dose and concomitant CRP concentrations were available. Approximately 6953 tacrolimus Cmin were excluded because of the absence of a concomitant CRP dosage, and then again 292 tacrolimus Cmin were excluded due to lack of data on the tacrolimus doses that were administered (see flow chart Figure 1). 

The demographic, pharmacological, and biological characteristics of the study population are presented in Table 1. The median age was 64 years old, and 82% of the subjects were men.

The median post-transplantation period during which the tacrolimus concentrations were measured was 14.2 months. The first quartile (Q1) of the delay post-transplantation was one month, which is in agreement with our inclusion criteria.

The median daily dose of tacrolimus was 3 mg and ranged from 3.5 to 14 mg, with a CV of 68%. The median Cmin of tacrolimus was 6.9 mg/L and ranged from 1 to 49.3 µg/L, with a coefficient of variation (CV) of 63.5%. 

Eighty-five percent of the measured tacrolimus Cmin were associated with liver enzyme concentrations within the normal range (*n* = 1150), corresponding to normal liver function.

The median level of CRP in the study population was 14 mg/L, corresponding to a low level of inflammation. CRP concentrations also showed high variability in our cohort with a CV of 134%.

### 3.2. Determinants of Tacrolimus Cmin Variability 

As the tacrolimus Cmin correlated with the tacrolimus dose (r = 0.285, *p* < 0.001), the analysis of the determinants of tacrolimus exposure variability was performed on the Cmin/Dose (C/D) to account for the influence of dose adjustments. The tacrolimus C/D ranged from 0.17 to 70, with a CV of 116%.

Univariate analysis showed the tacrolimus C/D to be significantly positively associated with the hematocrit, ALAT, CRP, and total bilirubin concentrations (Table 2). As the level of inflammation was low and very variable, we decided to consider CRP as a categorical variable by analyzing the quartiles: Q1 (range) mg/L: 4 (<3–4), Q2: 9 (5–14), Q3: 25 (15–46), and Q4: 88 (49–334). The hematocrit, ALAT, and CRP concentrations in the fourth quartile remained independent determinants of the tacrolimus C/D in multivariate analysis (Table 2). Post-transplantation delay was expressed as continuous variables or categorical variables (quartiles: Q1: (range) months): 1.90 (1.18–2.13), Q2: 4.57 (3.80–5.50), Q3: 11.62 (8.20–12.57), and Q4: 37.03 (22.54–56.87)) had no effect on log C/D.

### 3.3. Role of Inflammation in Tacrolimus Overexposure

We compared the CRP concentrations in patients with and without tacrolimus overexposure to further determine the influence of inflammation on cases of observed overexposure to tacrolimus (determined by tacrolimus Cmin > 15 µg/L). The CRP concentrations were higher (*p* = 0.003) for patients with tacrolimus overexposure (median CRP (10th–90th percentiles): 27 mg/L (3–149 mg/L), *n* = 91) than it was for patients with a tacrolimus Cmin ≤ 15 µg/L (13 mg/mL (3–95 mg/L), *n =* 1482)) (see Figure 2). Univariate analysis showed that tacrolimus overexposure was significantly associated with hematocrit, ASAT, total bilirubin, CRP, and the post-transplantation delay. In multivariate linear mixed-effect regression analysis, CRP in the fourth quartile, post-transplant delay (expressed either as continuous variables or quartiles) and ALAT concentrations remained independent determinants of tacrolimus Cmin overexposure (Table 3). However, the interaction between CRP quartiles and post-transplant delay had no significant effect on tacrolimus overexposure (*p* = 0.075). Similar results were obtained with the generalized logistic mixed model that showed a significant effect of the log ALAT (*X^2^* = 23.46, *p* < 0.001), CRP quartiles (*X^2^* = 7.61, *p* = 0.05), and log post-transplant delay (*X^2^* = 6.25, *p* = 0.012) on tacrolimus overexposure. Table 4 shows the odd ratios of these covariables on tacrolimus overexposure. Again, ALAT and, to a lesser extent, CRP is the fourth quartile that were identified as significant determinants of tacrolimus overexposure.

Modifications made to the dose following an episode of tacrolimus overexposure were only available for sixty-one samples (64% of the total overexposures). The tacrolimus dose had been changed in 67% of these overexposures (41/61). 

## 4. Discussion

This study provides evidence that inflammation contributes to tacrolimus Cmin and C/D variability.

We chose to study tacrolimus variability after the second hospitalization of liver transplant patients, i.e., at least 15 days after the graft, in order to allow liver function to recover, as liver function is a well-known major determinant of tacrolimus pharmacokinetic variability [7]. The finding that the median post-transplantation period was 14.2 months with a first quartile of 1.90 months confirmed that the Cmin of tacrolimus was not measured within the first month post-transplantation, which is when it is reported to be highly variable [7]. However, the tacrolimus Cmin still remained highly variable in our cohort, with a coefficient of variation of 63.5%. Since dose adjustments were performed during longitudinal TDM to maintain tacrolimus Cmin within the targeted therapeutic ranges, we choose to analyse the determinants of tacrolimus exposure variability on the C/D to account for the influence of dose adjustments.

Our results show that the ALAT, hematocrit, and CRP concentrations in the fourth quartile have a significant impact on tacrolimus C/D variability. 

The association between ALAT and tacrolimus C/D was expected, given that tacrolimus is highly metabolized by cytochrome P4503A4/3A5; decreased hepatic clearance could require reducing the daily dose to maintain the tacrolimus Cmin within the target therapeutic window, consequently leading to an increase in the tacrolimus C/D. In addition, tacrolimus is taken up by and binds to erythrocytes, resulting in a proportion of its related bound form increasing, along with an increase in hematocrit [23,24], which would explain the positive association between hematocrit and tacrolimus Cmin. In addition, acute inflammation episodes are characterized by the increased synthesis of acute-phase proteins, including alpha1-acid glycoprotein, for which tacrolimus shows high affinity and saturable binding capability [25]. Since the total form (bound and unbound forms) of tacrolimus was measured during TDM, the enhanced concentrations of tacrolimus that were observed during episodes of inflammation could mostly reflect the enhanced concentration of the bound (i.e., inactive) form of tacrolimus, as described for lopinavir, another drug with a high binding affinity to alpha1-acid glycoprotein [19,26,27]. Unfortunately, the alpha1-acid glycoprotein dosages were not available for most of the patients in our cohort, so we were not able to evaluate the relationship between tacrolimus C/D and alpha1 acid glycoprotein, meaning that this remains to be investigated in future studies.

Interestingly, in the present study, the relationship between the CRP concentrations and tacrolimus C/D was only significant for the fourth quartile of CRP. This finding can be explained by the normal to low inflammatory status of most of the patients in our cohort, with the exception of those in the fourth quartile, for whom the median CRP concentration was 88 mg/L, which corresponds to a medium to high level of inflammation. This finding suggests that a medium to high level of inflammation is required to induce an increase in tacrolimus C/D. This conclusion is consistent with that of a recent tacrolimus pharmacokinetics sub-study that was performed in kidney transplant patients who had also been treated with the anti-IL-6 therapeutic monoclonal antibody clazakinumab in the context of antibody-mediated rejection. In this randomized, double-blind, placebo-controlled phase 2 pilot trial, the authors reported that treatment with clazakinumab had no effect on the C/D of tacrolimus [28]. It should be noted that the underlying systemic inflammation was very low in this population of kidney transplant patients with antibody-mediated rejection (median CRP: 2 mg/L). Such as low grade inflammation had no significant effect on either the basal cytochrome P450-dependant metabolism [28] o on the ability of tacrolimus to bind to the plasma protein. 

Lastly, we found no statistically significant relationship between tacrolimus C/D and the post-transplant delay, which was probably because we chose to exclude the tacrolimus dosages that were administered during the first post-transplant hospitalization and during the first 15 days post-transplant from our study.

Collectively, our results suggest that changes in the tacrolimus Cmin measured in whole blood may partially reflect either decreased hepatic clearance, increased distribution within erythrocytes, or increased binding to alpha1 acid glycoprotein. Overall, these results are consistent with those of a recent study based on a Pharmacokinetic/Pharmacodynamic approach that identified the hematocrit, plasma unbound fraction, and intrinsic clearance as the main determinants of the tacrolimus Cmin in adult liver transplant patients [20]. 

Our second objective was to determine whether tacrolimus overexposure (Cmin > 15 µg/L) was associated with an enhanced inflammatory status. Our data showed that the concentrations of CRP were higher for patients with a tacrolimus Cmin > 15 µg/L than they were for those with a tacrolimus Cmin ≤ 15 µg/L, and that the concentrations of ALAT, CRP, and post-transplant delay were independent predictors of tacrolimus overexposure. 

The CRP concentration in the fourth quartile was associated with a 2.6-fold increased risk of tacrolimus overexposure. However, as tacrolimus is a drug with a low hepatic extraction ratio and a high binding affinity to plasma proteins, the inflammation-induced changes in the distribution processes are theoretically independent from the unbound drug concentration [29,30]. Thus, the dose of tacrolimus should not be systematically reduced when acute episodes of inflammation occur, even if the tacrolimus Cmin is > 15 µg/L. Conversely, decreased hepatic clearance that is consistent with liver failure or co-treatment with a cytochrome P450 inhibitor may require a reduction in the daily dose of tacrolimus to maintain the tacrolimus Cmin within the target therapeutic window. Our data show that the ALAT increase was associated with a 4-fold increased risk of tacrolimus overexposure and are in agreement with the impact of the reduced clearance of tacrolimus Cmin. Lastly, the risk of tacrolimus overexposure (Cmin > 15 µg/L) significantly decreased with the post-transplant delay. This finding could be explained by the fact that therapeutic targets are higher in the early post-transplant period (up to 10–15 µg/L) and are lower in later post-transplant periods (5–10 µg/L), depending on the immunosuppressive regimen that is prescribed [1]. 

All of these data highlight the strong need to have an overview of liver function, which is the best predictor of tacrolimus overexposure in our study, but also a need to have an overview of the inflammatory status and the post-transplant delay of patients treated with tacrolimus to correctly interpret any increase in their tacrolimus Cmin beyond co-medication with a cytochrome P450/3A4 inhibitor. 

We acknowledge that our study had several limitations. Its retrospective design did not allow us to collect data on the co-medications of patients nor did it allow us to genotype the CYP3A5 data although the drugs that inhibit cytochrome 3A4/3A5 or cytochrome P4503A/5 genetic polymorphisms highly contributed to tacrolimus Cmin variability [15]. In addition, few data were available to describe the adjustments of the dose of tacrolimus that was prescribed by the clinicians after an episode of tacrolimus overdosage. 

However, our data reflect real-life changes in the tacrolimus Cmin observed in a longitudinal patient follow-up, and pharmacogenetic data are rarely available for routine TDM in liver transplant patients. The size of our cohort of patients and the statistical methodology that was used, which took into account the within and between individual changes of tacrolimus concentrations and the doses and CRP concentrations that occurred during the longitudinal follow up, as well as the post-transplantation delays are strengths of our study.

In conclusion, our study suggests that inflammation should be taken into account for the correct comprehension and interpretation of the tacrolimus Cmin variability observed during longitudinal TDM that occurs separately from liver dysfunction, poor observance issues or blood exams performed at the wrong time, or co-medication with drug-metabolizing enzymes (cytochrome P4503A4/3A5) and transporter (P-glycoprotein) inducers or inhibitors. 

## Figures and Tables

**Figure 1 pharmaceutics-13-01960-f001:**
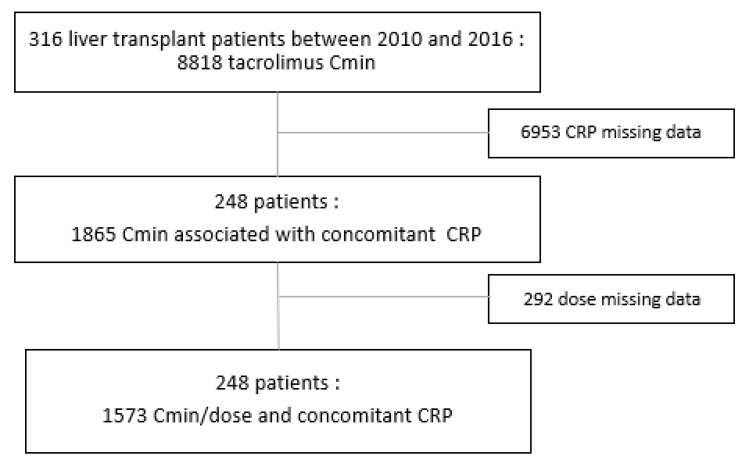
Flow chart of the study population.

**Figure 2 pharmaceutics-13-01960-f002:**
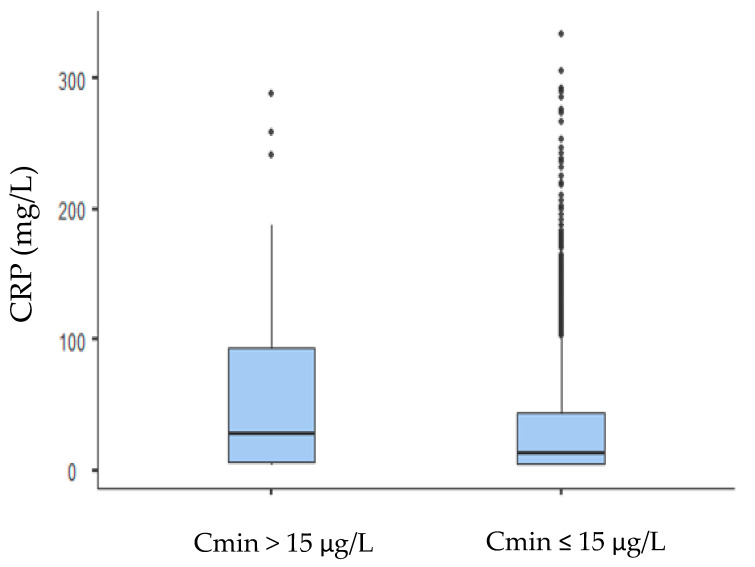
Relationships between tacrolimus blood exposure (normal range (<15 µg/L, *n* = 1482) or overexposure (>15 µg/L, *n* = 91) and CRP concentration. Data are presented as interquartile range (boxes), data range (whiskers), and median (horizontal line).

**Table 1 pharmaceutics-13-01960-t001:** Characteristics of patients and tacrolimus therapeutic drug monitoring.

Demographics (*n* = 248 Patients)	
Gender (male) % (*n*)	82 (202)
Age (years)	64 (51–72)
Duration of hospital stay (Days)	8 (0–76)
Number of recurrent hospitalizations	5 (2–17)
Post-transplant delays (months)	14.2 (1.60–45.2)
**Tacrolimus pharmacological data (*n* = 1573)**	
Concentration (µg/L)	6.9 (3.4–12.7)
Concentration/Dose	2.16 (0.92–6.47)
Dose (mg/Day)	3 (1–7)
Number of tacrolimus Cmin measurements/patient	4 (2–16)
**Laboratory parameters**	
CRP (mg/l) *n* = 1573	14 (3–100)
ASAT(UI/L) *n* = 1341	25 (10–97)
ALAT (UI/L) *n* = 1342	34 (16–149)
Total bilirubin (mg/L) *n* = 1445	11 (5–41)
Hematocrit (%) *n* = 1404	0.32 (0.25–0.42)
White blood cell count (G/L) *n* = 1404	5.4 (2.6–10.3)
Total protein (g/L) *n =* 1522	66 (51–77)

Data are presented as medians (10th–90th percentiles) or percentages (numbers).

**Table 2 pharmaceutics-13-01960-t002:** Univariate and multivariate linear mixed-effect regression analyses for the identification of the determinants of log tacrolimus trough concentrations adjusted for dose (*n* = 1573) during longitudinal therapeutic drug monitoring.

Variable	Univariate Analysis		Multivariate Analysis	
	Estimate ± SE	*p* Value	Estimate ± SE	*p* Value
Age	2. 55 ± 1.59 × 10^−3^	0.110	/	/
Gender			/	/
women	−0.05 ± 0.33	0.874	/	/
men	0.011 ± 0.33	0.966	/	/
Hematocrit	0.30 ± 0.16	0.049	0.51 ± 0.18	0.004
Log (ALAT)	0.10 ± 0.02	<0.001	0.10 ± 0.02	<0.001
Log (total bilirubin)	0.04 ± 0.02	0.060	0.01 ± 0.03	0.594
Log (post-transplant delay)	0.01 ± 0.02	0.382	/	/
Log (CRP)	0.05 ± 0.01	<0.001	/	/
1st–2nd quartile	4.32 ± 21.80 × 10^−3^	0.843	0.03 ± 0.02	0.159
3rd–1st quartile	3.00 ± 0.02	0.895	0.02 ± 0.03	0.510
4th–1st quartile	0.07 ± 0.02	0.003	0.10 ± 0.03	<0.001

CRP quartile median (range) mg/L: Q1 = 4 (<3–4), Q2 = 9 (5–14), Q3 = 25 (15–46), and Q4 = 88 (49–334).

**Table 3 pharmaceutics-13-01960-t003:** Univariate and multivariate linear mixed-effect regression analyses for the identification of the determinants of tacrolimus overexposure (tacrolimus trough concentrations > 15 µg/L (*n* = 91)) during longitudinal therapeutic drug monitoring.

Variable	Univariate Analysis		Multivariate Analysis	
	Estimate ± SE	*p* Value	Estimate ± SE	*p* Value
Hematocrit	0.23 ± 0.10	0.023	−0.08 ± 0.12	0.518
Log (ALAT)	−0.11 ± 0.01	<0.001	0.10 ± 0.02	<0.001
Age	8.90 ± 7.03 × 10^−3^	0.207	/	/
Gender				
women	−0.05 ± 0.23	0.841	/	/
men	−0.06 ± 0.23	0.800	/	/
Log (total bilirubin)	−0.09 ± 0.01	<0.001	0.04 ± 0.02	0.078
Log (CRP)	−0.03 ± 0.01	0.001	/	/
1st–2nd quartile	0.01 ± 0.01	0.526	−2.70 ± 17.51 × 10^−3^	0.878
3rd–1st quartile	0.2 ± 1.69 × 10^−3^	0.991	−0.12 ± 0.02 × 10^−3^	0.995
4th–1st quartile	−0.05 ± 0.02	0.005	−0.06 ± 0.02	0.005
Log (post-transplant delays)	0.04 ± 0.02	0.002		0.019
1st–2nd quartile	−0.02 ± 0.02	0.140	−0.05 ± 0.01	0.029
3rd–1st quartile	−0.06 ± 0.02	<0.001	−0.06 ± 0.02	0.003
4th–1st quartile	−0.04 ± 0.02	0.013	−0.05 ± 0.02	0.025

**Table 4 pharmaceutics-13-01960-t004:** Generalized logistic mixed model for the identification of determinants of tacrolimus overexposure.

Variable	Odd Ratio	IC 95%	*p* Value
Log (ALAT)	4.10	2.32–7.27	<0.001
log (total bilirubin)	1.61	0.85–3.08	0.146
Hematocrit	0.23	0.13–40.25	0.577
CRP			
1st–2nd quartile	1.09	0.46–2.55	0.846
3rd–1st quartile	1.22	0.50–2.98	0.656
4th–1st quartile	2.64	1.13–6.17	0.024
Log (Post-transplant delay)	0.72	0.56–0.93	0.012

## Data Availability

Data available on request to the corresponding author.

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
