# Peer review of "Variability of Tacrolimus Trough Concentration in Liver Transplant Patients: Which Role of Inflammation?"

_pharmaceutics, 2021, doi:10.3390/pharmaceutics13111960_

Round 1

Reviewer 1 Report

The research work is very well written and justified through suitable evaluation parameters and references. Though it contains sufficient novelty to be accepted for publication, but still minor modifications and suggestions are recommended to improve the quality of the manuscript.

  • Materials and Methods:
  • Line 67. What drug containing tacrolimus was administered to patients? Add the name of preparation, producer.
  • Have patients been ordered to receive tacrolimus with another immunosuppressive drug (or steroids immunosuppression)? If yes, complete the data.
  • Line 77. Describe the procedure for determination of: CRP, alanine aminotrasferase (ALAT), aspartate aminotransferase (ASAT), haematocrit, protein, bilirubin, tacrolimus Cmin. Specify methods of determination, reagents used (purity grade, producer, country).
  • Line 81. Describe the test procedure by a liquid chromatography tandem mass spectrometry method. Specify the reagents used (purity grade, producer, country).
  • Results:
  • Figure 2: The underlining in red of Cmin is unnecessary.
  • Units should be standardised throughout the article (e.g. CRP: mg/L or mg/l).
  • Table 2, 3, 4. No analysis of ASAT results.

Conclusion
Overall, the manuscript could be considered as scientific rigor and seems able to add in existing scientific knowledge. Therefore, I recommend the Acceptance of the manuscript with minor modifications on above mentioned suggestions and comments.

Author Response

Responses to reviewer 1
The research work is very well written and justified through suitable evaluation parameters and references. Though it contains sufficient novelty to be accepted for publication, but still minor modifications and suggestions are recommended to improve the quality of the manuscript.

The authors thank the referee for his/her positive comments.

Materials and Methods:
Line 67. What drug containing tacrolimus was administered to patients? Add the name of preparation, producer.
Patients were treated with either Prograf, Advagraf or Envarsus. This point is now included in the revised version.

Have patients been ordered to receive tacrolimus with another immunosuppressive drug (or steroids immunosuppression)? If yes, complete the data.
All patients also received corticosteroids and mycophenolate mofetil for the prevention of graft rejection. This point is now included in the revised version.

Line 77. Describe the procedure for determination of: CRP, alanine aminotrasferase (ALAT), aspartate aminotransferase (ASAT), haematocrit, protein, bilirubin, tacrolimus Cmin. Specify methods of determination, reagents used (purity grade, producer, country).
The analytical procedure for CRP, alanine aminotrasferase (ALAT), aspartate aminotransferase (ASAT), haematocrit, protein, bilirubin, and tacrolimus Cmin are described in an additional paragraph entitled “analytical methods” in the revised version.

Line 81. Describe the test procedure by a liquid chromatography tandem mass spectrometry method. Specify the reagents used (purity grade, producer, country).
The LC-MS method is now described in the revised version in an additional paragraph “analytical method”.

Results:
Figure 2: The underlining in red of Cmin is unnecessary.
The underlining in red of Cmin has been corrected in the revised version.

Units should be standardised throughout the article (e.g. CRP: mg/L or mg/l).
Table 2, 3, 4. No analysis of ASAT results.
ASAT was not included in the analysis because of its collinearity with the ALAT variable. We chose to include ALAT variable in the multivariate analysis, since ALAT is more specifically related to liver dysfunction. This point is now indicated in the revised version (statistical analysis section).

Conclusion
Overall, the manuscript could be considered as scientific rigor and seems able to add in existing scientific knowledge. Therefore, I recommend the Acceptance of the manuscript with minor modifications on above mentioned suggestions and comments.

Reviewer 2 Report

The topic of this manuscript is interesting and fits well the scope of Pharmaceutics. The reviewer feels it can be accepted after some minor amendments.

(1) The language should be polished. 

What does "Thus, the identification and consideration of factors involved
 in tacrolimus Cmin variability is highly clinically important to prevent or limit it."    "... to prevent or limit it." Very odd. 

(2) Why used the log Cmin/dose of tacrolimus instead of absolute value of Cmin/dose?

(3) The LC-MS/MS method should be described. 

Author Response

Reviewer 2 :

The topic of this manuscript is interesting and fits well the scope of Pharmaceutics. The reviewer feels it can be accepted after some minor amendments.

(1) The language should be polished. 

The manuscript has been corrected by a native English speaker before being spent.

What does "Thus, the identification and consideration of factors involved
in tacrolimus Cmin variability is highly clinically important to prevent or limit it." "... to prevent or limit it." Very odd. 
The sentence has been corrected according to the referee’s comment.

(2) Why used the log Cmin/dose of tacrolimus instead of absolute value of Cmin/dose?

As indicated in the statistical analysis section, data were log-transformed to satisfy the conditions of application of linear models when not normally distributed.  That was the case for Cmin/Dose. 
Log Cmin/dose was therefore used instead of absolute value of Cmin/dose in order to satisfy the conditions of application of linear models when the variable does not follow a normal distribution. 

(3) The LC-MS/MS method should be described.
The LC-MS/MS method is now described in the revised version.

Reviewer 3 Report

The authors should consider the followings:

  1. In the method validation of LCMS of Tacrolimus in blood, did the authors establish the interference levels with bilirubin, lipedemic, and/or hemolyzed blood samples?
  2. What was the monitored concentration level of low QC? and what was the CV of the LLOQ?
  3. Please provide the ethnicity of the demographics
  4. Please elaborate why the authors chosen the analysis without the control subjects.
  5. The authors should list the co-medication with drug metabolizing enzymes (cytochrome P4503A4/3A5) and transporter (P-glycoprotein) inducers or inhibitors, of the subjects studied.
  6. In Table 1, the authors may provide several cases of patients, with their tacrolimus Cmin measurements across the time-line. (as supplementary information, if available)
  7. The authors may modify the title of the article to better reflect the results of the study.
  8. How long would the samples be stored before the start of the sample analysis? Would the duration and temperature of the temporary storage of the sample being monitored and complied by the stability results?
  9. In Figure 2, please justify the sample numbers of those doses of the two groups in comparison. Please list the total number of CRP records included in each group.
  10. The authors may supplement the methodology of LCMS platform used in the study.
  11. "few data were available to describe the adjustments of the dose of tacrolimus that were performed by the clinicians after an episode of tacrolimus overdosage" The authors should further elaborate this statement. Did the author systematically request or collect each of the tacrolimus overdosage episodes?
  12. The authors should be more specific in your abstract and conclusion that, indicating which results of your study was found to be novel.

Author Response

Response to Reviewer 3
The authors should consider the followings:
1.In the method validation of LCMS of Tacrolimus in blood, did the authors establish the interference levels with bilirubin, lipedemic, and/or hemolyzed blood samples?
In the method validation of tacrolimus quantitatition by a LC/MS method, the authors did not established the interference levels with bilirubin, lipemic and/or hemolysed samples since these analytical issues were fixed given the use of 13C-D2-tacrolimus tacrolimus which underwent the same potential sample-dependent analytical interferences than tacrolimus. The method used is very common and has been used for almost 20 years in many clinical centers. Our laboratory is enrolled in a national and bi-monthly quality control program that consisted in blind analyses of pools of whole blood taken from patients treated with tacrolimus, and our results were always in line with expected results.

2.What was the monitored concentration level of low QC? and what was the CV of the LLOQ?
The lower level of QC was 2.42 µg/L. 
The between-day CV of the LLOQ (1µg/L) was 9.64%.
The levels of QC and their corresponding between-day CV are now indicated in the revised version as well as the CV of the LLOQ.

3.Please provide the ethnicity of the demographics
The ethnicity of the demographic is not registered in the medical file, so these data were not available. 
4.Please elaborate why the authors chosen the analysis without the control subjects.
Our aim was to determine the contribution of inflammation in the variability of tacrolimus Cmin in a large cohort of liver transplant patients. The design of our study therefore implies the absence of control subjects.

5.The authors should list the co-medication with drug metabolizing enzymes (cytochrome P4503A4/3A5) and transporter (P-glycoprotein) inducers or inhibitors, of the subjects studied.
The referee is correct to mention that the list of co-medication with drug metabolizing enzymes (cytochrome P4503A4/3A5) and transporter (P-glycoprotein) inducers or inhibitors would have been interesting. However, as indicated in the discussion section (limitations of the studies), these data were not available. We acknowledged that this point was a limitation of our study in the first version. 

6.In Table 1, the authors may provide several cases of patients, with their tacrolimus Cmin measurements across the time-line. (as supplementary information, if available)
Our aim was to determine the contribution of inflammation in the variability of tacrolimus Cmin in a large cohort of liver transplant patients. We have already published variations of tacrolimus Cmin measurements across the time-line in liver transplant patients in relation to acute episode of inflammation (see Bonneville et al, Br J Clin Pharmacol 2020). The present study was precisely of continuation of this later study.

7.The authors may modify the title of the article to better reflect the results of the study.
The title has been changed as “Variability of tacrolimus trough concentration in liver transplant patients: which role of inflammation?”

8.How long would the samples be stored before the start of the sample analysis? Would the duration and temperature of the temporary storage of the sample being monitored and complied by the stability results?
This is a retrospective study on data collected on medical records. So all analyses were performed during daily routine care on fresh samples.

9.In Figure 2, please justify the sample numbers of those doses of the two groups in comparison. Please list the total number of CRP records included in each group.
Table 2 presents the relationships between CRP concentration and tacrolimus blood exposure within the normal range (<15 µg/L, n = 1482) vs tacrolimus overexposure (> 15 µg/L, n = 91). 
The number of samples is now indicated in the revised version. 

10.The authors may supplement the methodology of LCMS platform used in the study.
The LC-MS/MS method is now described in the revised version.

11."few data were available to describe the adjustments of the dose of tacrolimus that were performed by the clinicians after an episode of tacrolimus overdosage" The authors should further elaborate this statement. Did the author systematically request or collect each of the tacrolimus overdosage episodes?

All overdose episodes were systematically collected but for only 64% of them the administered dose was available to evaluate the adjustment of the dose, as mentioned in the result section. We acknowledged that was a limitation of our study.

12.The authors should be more specific in your abstract and conclusion that, indicating which results of your study was found to be novel.
The abstract has been changed, keeping in mind the authorized number of words. The following sentence has been added in the revised version. “CRP in the fourth quartile (49 to 334 mg/L) was associated with a 2.6-fold increased risk of tacrolimus Cmin overexposure. “

This manuscript is a resubmission of an earlier submission. The following is a list of the peer review reports and author responses from that submission.